# Manganese Overexposure Alters Neurogranin Expression and Causes Behavioral Deficits in Larval Zebrafish

**DOI:** 10.3390/ijms25094933

**Published:** 2024-04-30

**Authors:** Anabel Alba-González, Elena I. Dragomir, Golsana Haghdousti, Julián Yáñez, Chris Dadswell, Ramón González-Méndez, Stephen W. Wilson, Karin Tuschl, Mónica Folgueira

**Affiliations:** 1Department of Biology, Faculty of Sciences, University of A Coruña, 15008 A Coruña, Spain; anabel.albag@udc.es (A.A.-G.); julian.yanez@udc.es (J.Y.); 2Centro Interdisciplinar de Química y Biología, (CICA), University of A Coruña, 15071 A Coruña, Spain; 3Department of Cell and Developmental, University College London, London, WC1E 6BT, UK; e.dragomir@ucl.ac.uk (E.I.D.); golsana.haghdousti.20@ucl.ac.uk (G.H.); s.wilson@ucl.ac.uk (S.W.W.); 4School of Life Sciences, University of Sussex, Brighton, BN1 9QJ, UK; c.m.dadswell@sussex.ac.uk (C.D.); r.gonzalez-mendez@sussex.ac.uk (R.G.-M.); 5UCL GOSH Institute of Child Health, University College London, London, WC1N 1EH, UK

**Keywords:** *Danio rerio*, neurotoxicity, neurogranin, manganese, development

## Abstract

Manganese (Mn), a cofactor for various enzyme classes, is an essential trace metal for all organisms. However, overexposure to Mn causes neurotoxicity. Here, we evaluated the effects of exposure to Mn chloride (MnCl_2_) on viability, morphology, synapse function (based on neurogranin expression) and behavior of zebrafish larvae. MnCl_2_ exposure from 2.5 h post fertilization led to reduced survival (60%) at 5 days post fertilization. Phenotypical changes affected body length, eye and olfactory organ size, and visual background adaptation. This was accompanied by a decrease in both the fluorescence intensity of neurogranin immunostaining and expression levels of the neurogranin-encoding genes *nrgna* and *nrgnb*, suggesting the presence of synaptic alterations. Furthermore, overexposure to MnCl_2_ resulted in larvae exhibiting postural defects, reduction in motor activity and impaired preference for light environments. Following the removal of MnCl_2_ from the fish water, zebrafish larvae recovered their pigmentation pattern and normalized their locomotor behavior, indicating that some aspects of Mn neurotoxicity are reversible. In summary, our results demonstrate that Mn overexposure leads to pronounced morphological alterations, changes in neurogranin expression and behavioral impairments in zebrafish larvae.

## 1. Introduction

Manganese (Mn) is an essential trace metal that is required for many aspects of cell physiology. It acts as a cofactor for numerous enzymes (such as glutamine synthetase, arginase and superoxidase dismutase) and is thereby involved in neurotransmitter signaling; immune function; and carbohydrate, vitamin and energy metabolism [1,2,3]. Tight regulation of Mn homeostasis is essential for biological functions, with the intestine and liver playing key roles in its homeostatic control [4,5,6,7,8,9]. Despite being essential, high levels of Mn are neurotoxic since Mn accumulates in the brain, particularly the basal ganglia, cerebellum and frontal cortex [10,11,12,13,14]. Mn overload can occur due to occupational and environmental overexposure, impaired excretion due to liver dysfunction or inherited Mn transporter defects caused by mutations in SLC30A10 and SLC39A14 [7,13]. Mn enters the brain via the blood–brain barrier (using various transporter proteins) or reaches the central nervous system via the olfactory system [12,15,16]. 

Mn accumulates in the brain in a dose-dependent manner [17,18,19,20,21]. At high concentrations, it can be transported into both pre- and post-synaptic neurons via voltage-dependent Ca^2+^ channels [22,23,24,25]. Its accumulation has been suggested to decrease the levels of Gamma-aminobutyric acid (GABA), glutamate and dopamine, thereby affecting synaptic neurotransmission [24,26,27]. In humans, Mn overexposure causes manganism, an extrapyramidal movement disorder associated with psychiatric symptoms [7,12,17,28,29,30]. Furthermore, Mn dyshomeostasis is a feature of common neurodegenerative disorders, such as Alzheimer’s and Parkinson’s diseases [31,32,33]. Vertebrate animals, including guinea pigs [34], mice [35] and zebrafish [36,37,38], have been used to study the mechanisms of Mn neurotoxicity since they share cellular or whole organism phenotypes, including postural behavioral impairments upon Mn exposure.

Biomarkers of neurodegeneration and neuronal injury include altered expression of many synaptic proteins, including the calmodulin-binding protein neurogranin (NRGN) [39,40,41,42,43,44]. In humans, neurodegenerative diseases such as Alzheimer’s disease lead to a depletion of NRGN expression in the brain and an elevation of NRGN levels in the cerebrospinal fluid, changes that are associated with diminished cognitive performance [45,46,47]. Here, we have evaluated the acute toxicity effects of MnCl_2_ on zebrafish larvae, including potential effects on synapses based on Nrgn expression [48,49] as well as larval exploratory behavior, light/dark preference and optomotor response.

## 2. Results

### 2.1. MnCl_2_ Exposure Leads to Reduced Survival of Zebrafish Larvae

To determine a suitable concentration for studying the effects of MnCl_2_ toxicity in zebrafish development, embryos were exposed to increasing concentrations of MnCl_2_ from 2.5 h post-fertilization (hpf), and survival was analyzed at 5 days post fertilization (dpf) (Appendix A). Survival was 100% up until 4 dpf (Appendix A). By 5 dpf, concentrations above 200 µM increased mortality compared to control larvae (200 µM: 94% survival; 300 µM: 80% survival), with 500 µM MnCl_2_ reducing survival to 64% (Appendix A). 

### 2.2. MnCl_2_ Exposure Affects Body Length, Eye Size and Olfactory Organ Morphology 

Given the decrease in survival after exposure to 500 µM MnCl_2_, we determined the effects of this concentration on larval morphology (Figure 1a–d). MnCl_2_ exposure (500 µM) from 2.5 hpf to 5 dpf caused a significant reduction in body length, while shorter exposure durations did not affect this parameter (Figure 1a(I–IV),b; Appendix A). This was accompanied by an increase in eye size relative to body length as well as a decrease in olfactory organ size (Figure 1c,d; Appendix A). The zebrafish olfactory epithelium contains different cell types, including ciliated and microvillous sensory cells [50,51]. Tubulin immunohistochemistry, which labels the cytoskeleton, as well as scanning electron microscope analysis, suggested that MnCl_2_ exposure causes structural changes in the olfactory organ, suggesting an increase in the length of the apical processes of sensory cells of the olfactory epithelium (Figure 2a–d). 

Given that we observed a reduction in the size of the olfactory epithelium after exposure to 500 µM MnCl_2_, we addressed whether this exposure affected the area of the lateral profile of the intact brain or, more specifically, of telencephalic regions that receive olfactory afferents (telencephalon, olfactory bulb and telencephalic lobes). Although the size of the olfactory organ was altered, the area of the lateral profile of the olfactory bulb, the telencephalic region that receives the afferents from the olfactory epithelium [50,51], was unchanged. We did not observe differences in the size of the area of the lateral profiles of the whole brain nor the telencephalon between exposed and untreated larvae (Figure 3).

### 2.3. MnCl_2_ Toxicity Leads to Reduced Intensity of Neurogranin Immunoreaction and mRNA Expression

Although the area (in lateral views) of the brain and telencephalon was not overtly changed upon MnCl_2_ exposure (500 µM from 2.5 hpf to 5 dpf), it led to reduced Nrgn protein expression at 5 dpf, reflected as a decrease in mean fluorescent intensity in the whole brain of zebrafish larvae (Figure 4a,b,d). MnCl_2_ exposure before 2 dpf did not lead to any obvious changes in Nrgn expression in any of the regions analyzed (Figure 4a,c,d), suggesting that sensitivity to MnCl_2_ depends on the duration of exposure and/or timepoint within embryonal development. To assess whether MnCl_2_ exposure affects the transcriptional regulation of *nrgn* and protein abundance, mRNA levels of the two zebrafish paralogs, *nrgna* and *nrgnb*, were assessed by qRT-PCR. Indeed, MnCl_2_ exposure led to reduced mRNA expression of both *nrgna* and *nrgnb* at 4 dpf (Figure 4e). 

### 2.4. MnCl_2_ Exposure Leads to Mn Accumulation and Alters Larval Zebrafish Behavior 

MnCl_2_ neurotoxicity has previously been linked to impaired locomotion in zebrafish larvae [52,53]. To further ascertain behavioral alterations in response to MnCl_2_ exposure, we subjected zebrafish larvae to behavioral analysis at 6 dpf (Appendix A). For these experiments, we reduced the MnCl_2_ concentration to 100 µM, which did not affect survival (Appendix A). Concentrations above 100 µM MnCl_2_ resulted in severe locomotion impairment and lack of swim bladder inflation (Appendix A) and were therefore unsuitable for assessing behavioral alterations. Mn accumulation was confirmed in 6 dpf larvae treated with 100 µM using ICP-MS for the exposure periods applied during behavioral testing (Figure 5). Whole larval Mn levels increased with exposure duration. Removal of MnCl_2_ from 4 dpf resulted in near normalization of Mn levels at 6 dpf (Figure 5a, Appendix A). 

Mn accumulation in larvae treated with 100 µM MnCl_2_ was accompanied by changes in zebrafish pigmentation with melanophore dispersion, also known as absent visual background adaptation (Figure 5b,c and Appendix A). Similar changes in pigmentation patterns have been described in zebrafish carrying loss-of-function mutations in the Mn transporter Slc39a14 that lead to Mn overload [30]. Removal of MnCl_2_ from 4 dpf led to normalization of the pigmentation pattern, suggesting that the Mn level directly correlates with melanophore changes (Figure 5b,c). 

We observed that 100 µM MnCl_2_ exposure resulted in postural deficiencies characterized by abnormal swimming patterns. Analysis of different exposure durations suggested that these postural deficiencies were more enhanced in larvae with longer exposure time to MnCl_2_ (Appendix A). Larvae exposed from 2 to 4 dpf that showed recovery of pigmentation at 6 dpf were also able to recover their locomotor behavior, suggesting that at least some Mn toxicity effects are reversible (Figure 5b; Appendix A).

Locomotor behavior analysis demonstrated that during exploration, MnCl_2_-treated larvae initiated fewer swim bouts both in light and darkness (Figure 6a). These differences were enhanced upon longer periods of MnCl_2_ exposure. Depending on illumination, unexposed larvae modulated their locomotion and tended to move less in the dark, in agreement with previous studies [54,55], but MnCl_2_-exposed larvae did not show this modulation (Figure 6a). Behavioral analysis suggested some differences in other locomotor parameters, such as mean bout velocity and mean bout displacement; however, these were not statistically significant (Figure 6b,c; Appendix A).

To assess whether MnCl_2_-exposed larvae show any deficits in goal-oriented navigation, as well as potential alterations in anxiety-related behaviors, we subjected larvae to a light/dark preference assay. Zebrafish larvae show positive phototaxis as they are attracted to illuminated areas, rapidly orient themselves and navigate toward the light source [56]. Additionally, dark avoidance behavior has been previously used to measure anxiety-related phenotypes [57,58]. While at 6 dpf, unexposed fish rapidly swam to the illuminated half field of the behavioral area and stayed there for the remainder of the assay (5 min); fish exposed to MnCl_2_ from 2 to 6 dpf and 3 to 6 dpf showed a reduction in the fraction of time spent in the illuminated half field (Figure 6d). This deficit was more pronounced upon longer exposure duration from 2 to 6 dpf. Mn 2–4 dpf exposed fish showed similar behavior to unexposed larvae at 6 dpf (Figure 6d), indicating that the behavioral deficits are either reversible following MnCl_2_ removal or that Mn accumulation in this protocol is insufficient to elicit toxicity.

To test whether MnCl_2_ exposure affects visual function as previously observed in zebrafish that carry loss-of-function mutations in Slc39a14 [30], we examined the optomotor response (OMR), a stabilization reflex in which fish orient themselves and swim in the direction of perceived whole-field visual motion [59]. When subjected to leftwards and rightwards whole-field motion, both MnCl_2_-exposed and unexposed fish modulated their turning behavior according to the perceived visual motion direction, indicating that MnCl_2_ exposure did not result in gross visual deficits (Figure 6e, Appendix A). 

In summary, MnCl_2_ exposure results in locomotor deficits characterized by impaired posture, reduced bout initiations, altered bout dynamics, and impaired light preference/dark avoidance, especially following longer periods of Mn exposure, suggesting a possible anxiety phenotype. The visual function underlying the OMR appears to be preserved.

## 3. Discussion

Given the continuous increase in environmental pollution as a result of human activities, the effects of Mn overexposure on human health and animal welfare are an important public health concern [60,61]. Our study demonstrates that MnCl_2_ overexposure in zebrafish larvae produces a robust disease model for the study of Mn neurotoxicity, with exposed larvae showing distinct morphological and behavioral deficits. Synaptic alterations suggested by reduced neurogranin expression further link Mn overexposure with neuronal toxicity. Our results underpin the importance of tight homeostatic control of Mn to preserve neuronal function. 

### 3.1. Mn Overexposure Induces Morphological Changes in Zebrafish Larvae

Our results confirm that MnCl_2_ exposure during zebrafish embryonic and larval development affects survival, with only ~60% of larvae exposed to 500 µM surviving beyond 5 dpf, which is in agreement with previous observations [38,62]. Mn accumulation and toxicity depend not only on MnCl_2_ concentration but also on the duration and developmental time point of exposure. Early developmental stages appear to be particularly sensitive to Mn toxicity, leading to defects in brain development, neuronal function and behavior (this study and [63,64,65]). 

Our study reveals that MnCl_2_ exposure results in shorter body length and altered eye and olfactory organ size in zebrafish larvae. Shorter body length upon Mn exposure has previously been observed in other organisms such as domestic animals, swine, poultry and mice [66,67,68]. 

Metals are long known to act as olfactory toxicants. Mn^2+^ can be taken up by primary olfactory neurons [69,70,71,72,73], gaining access to the brain via this route. We observed a reduction in the size of the olfactory epithelium during all periods of exposure to MnCl_2_. This reduction does not seem to occur as a result of disturbed organogenesis, as exposure to MnCl_2_ from 1.5 dpf to 3.5 dpf, when the olfactory epithelium is already differentiated [74], also results in smaller olfactory epithelium size. Thus, size reduction might be a consequence of cell death and/or decreased cell renewal, as shown for other metals [75,76,77]. The different cell types in the olfactory epithelium show variations in sensitivity to metal toxicity, with ciliated being more sensitive than microvillous olfactory sensory neurons [77,78,79,80,81]. It remains to be determined whether this is also the case for Mn. Our results suggest that there are morphological alterations affecting the apical processes of the olfactory epithelium after exposure to MnCl_2_, similar to that of copper and zinc toxicity [77,78]. Zebrafish neuromasts show structural changes in their apical domain after Mn exposure, which results in altered function [36]. Whether the structural alterations in the olfactory epithelium result in functional alterations will be the subject of future studies. 

### 3.2. Neurogranin Protein and Gene Expression Is Diminished upon MnCl_2_ Exposure

Neurogranin is a biomarker for synaptic dysfunction in neurodegenerative disorders [39,40,41,42,43,44], and our work demonstrates that MnCl_2_ exposure reduces neurogranin protein immunoreactivity. This is associated with transcriptional downregulation of both *nrgna* and *nrgnb* expression. This is consistent with other studies showing alteration of synapse-related proteins as a result of Mn exposure. For instance, changes in mRNA levels of the synaptic adhesion protein *neurexin 2a* have previously been linked to Mn neurotoxicity [52]. NRGN is a calmodulin-binding protein whose phosphorylation depends on intracellular Ca^2+^ levels [82,83,84]. NRGN phosphorylation leads to calmodulin release, favoring long-term potentiation over long-term depression [84,85,86]. Thus, Nrgn reduction as a result of Mn overexposure in zebrafish may affect other synapse-related proteins, including glutamate receptors, RNA binding proteins, or selective ion channels, among others [85,86,87,88,89], ultimately affecting the regulation of synaptic plasticity. Mn and other metals interfere with Ca^2+^ homeostasis [13,90], which may secondarily affect Nrgn expression. 

We demonstrate that some neurotoxic effects of Mn at early larval zebrafish stages are reversible. This would suggest that Mn causes synaptic dysfunction that precedes neurodegeneration. Previous studies show increased apoptosis in both neuronal cell cultures [91] and in the zebrafish nervous system [38,53] upon Mn toxicity. However, a MnCl_2_ concentration comparable to that used in our study did not show changes in the number of apoptotic cells in *slc39a14^−/−^* zebrafish [13]. This further suggests that MnCl_2_ neurotoxicity produces deficits in neuronal function prior to leading to neurodegeneration. This would also be consistent with locomotor deficits in MnCl_2_-exposed larvae being reversible to a great extent when MnCl_2_ is removed from the fishwater. 

### 3.3. MnCl_2_ Exposure Produces Postural Defects and Behavioral Impairment in Zebrafish Larvae

We observe that MnCl_2_ exposure in larval zebrafish results in postural defects, reduced motor activity and abnormal swim patterns both in light and dark conditions. Our results are in agreement with previous results [36,53], also showing a reduction in distance traveled and alterations in swimming patterns. Although not assayed in our study, postural instability in fish embryos after Mn exposure has been linked to neuromast malfunction, showing not only structural but also functional alterations in hair cells after Mn overexposure [36]. In addition, postural stability is closely linked to normal otolith development. Other metals, such as cadmium, have been shown to interfere with otolith development through interference with calcium physiology, thereby causing impaired balance control and swimming activity in larval zebrafish [92]. 

In addition to postural defects, MnCl_2_-exposed larvae demonstrated reduced motor output characterized by fewer bout initiations. Longer periods of MnCl_2_ exposure resulted in a gradual decrease in the number of bouts and, consequently, in the distance traveled both in light and dark conditions. The behavioral modulation between light and dark environments observed in wild-type fish was not as pronounced as in Mn-treated fish, indicating that they are less adaptive to changes in the environment. Furthermore, MnCl_2_-exposed larvae showed a reduced light preference and dark avoidance behavior, as observed in the light/dark choice assay. This is unlikely to be due to a reduction in the ability of the fish to detect illumination differences, as treated fish did not show significant deficits in OMR behavior despite the observed alteration in eye size and abnormal visual background adaptation. While the altered behavior could reflect impaired anxiety regulation [57], a generalized decreased response to environmental stimuli could also underly this phenotype, also supported by the lack of locomotor modulation after the dark environment transition. While there is evidence that manganese toxicity affects multiple neurotransmitter systems such as dopaminergic, glutamatergic and GABAergic signaling [93,94,95,96,97,98,99], the mechanisms of how manganese interferes with neurotransmitter signaling remain to be determined. 

In summary, our work demonstrates that MnCl_2_ exposure in zebrafish larvae leads to Mn neurotoxicity that is characterized by distinct morphological, locomotor and synaptic protein changes, thereby providing a disease model that allows further study of the mechanisms underlying Mn-related neuronal biology. 

## 4. Materials and Methods

### 4.1. Animal Maintenance and Embryo Collection

Wild-type zebrafish (*Danio rerio*; TL and AB strains) were kept under standard conditions Aleström et al. [100]. Briefly, they were fed twice a day with commercial dry flake food (TL line: JBL NovoBel and Sera Vipan) and *Artemia* sp. Fish were kept in aquaria at 28.0 ± 1.0 °C and under 14/10 h light/dark periods. System water parameters (pH, conductivity, nitrate, nitrite and ammonium) were monitored according to Aleström et al. [100]. For breeding, two females and one male fish were transferred to mating tanks. Embryos and larvae at a density of 50 larvae per 25 mL were maintained in Petri dishes with sterile dechlorinated tap water (pH = 7.4 ± 0.3; conductivity = 730 ± 30 µS) at 28.0 ± 1.0 °C and under 14/10 h light/dark periods.

All experimental procedures were conducted under license from the United Kingdom Home Office, following the United Kingdom Home Office regulations and/or European Community Guidelines on animal care and experimentation, and were approved by the animal care and use committees. 

### 4.2. MnCl_2_ Exposure

Zebrafish embryos were transferred to 4-well plates (10–12 embryos per well) and exposed to manganese chloride tetrahydrate (MnCl_2_; Sigma Aldrich, M-3634, St. Louis, MI, USA) in sterile dechlorinated tap water at concentrations ranging from 100 to 500 µM on a semistatic experiment (medium replaced every 24 h). Negative controls were maintained in sterile dechlorinated tap water, which was also replaced every 24 h. Various exposure periods were analyzed (see Appendix A).

### 4.3. Survival Analysis

Survival analysis was performed by comparing untreated larvae with larvae exposed to different concentrations of MnCl_2_ (200, 300 and 500 µM). Untreated larvae were raised in sterile dechlorinated water without the addition of MnCl_2_. Fish were monitored every 24 h up to 5 dpf, noting the presence of any of the five endpoints included in the OECD guidelines (Zebrafish Embryo Toxicity Test N° 236; OECD [101]). Briefly, individuals were analyzed for the following: (i) presence of edema; (ii) non-detachment of the tail from the yolk sac; (iii) absence of somite formation; (iv) lack of heartbeat after 2 dpf; or (v) coagulation of fertilized eggs. In addition, survival to 100 µM MnCl_2_ was monitored after exposure to the same periods analyzed in the behavior studies; that is, from 2 to 4 dpf (Mn 2–4 dpf), from 3 to 5 dpf (Mn 3–5 dpf) and from 2 to 5 dpf (Mn 2–5 dpf). 

### 4.4. Morphological Analyses

Zebrafish embryos were exposed to MnCl_2_ (500 µM) from 2.5 hpf to 5 dpf (Mn 2.5 hpf–5 dpf), from 1.5 dpf to 3.5 dpf (Mn 1.5 dpf–3.5 dpf) and from 2.5 hpf to 2 dpf (Mn 2.5 hpf–2 dpf). Seven parameters were analyzed at 5 dpf, including standard body length (µm), whole brain size (µm^2^), and normalized size values (structure size to body length ratio) of eyes and olfactory epithelium organ (diameter), the telencephalic hemisphere surface (from lateral view), olfactory bulbs and the telencephalic lobes surface area (from lateral view). 

Body length and eye diameter size were analyzed from images taken using a brightfield microscope (Nikon Eclipse 90i) coupled to an Olympus DP71 digital camera. Body length was considered as the distance from the mouth to the start of the caudal fin (µm). To measure the eyes and olfactory organ size, we measured the diameter normalized to the body length. The eyes were measured in lateral view, while the olfactory organs were in dorsal view. To determine brain area, we measured the area in lateral view using the obex as the caudal limit. To determine the area of the telencephalon and its subdivisions (the olfactory organ, the telencephalic hemisphere (measured from the anterior edge of the olfactory bulb to the caudal end of the telencephalic lobes), the olfactory bulbs and the telencephalic lobes), tubulin immunofluorescence technique in whole-mounts was used (see below), and measurements were performed in lateral views. 

### 4.5. Immunohistochemistry

To determine effects on the brain and olfactory organ size and variations in neurogranin expression after MnCl_2_ exposure, tubulin and neurogranin immunoreaction were analyzed, respectively, following the protocol described by Turner et al. [102]. Briefly, larvae were euthanized with tricaine methane sulfonate 0.3 mg/mL (MS222; Sigma, St. Louis, MO, USA) and fixed in sweet 4% paraformaldehyde (PFA) in 0.1 M phosphate buffer (pH 7.4) overnight at 4 °C. If required, brains were exposed by manually removing skin, eyes and bone under a stereomicroscope [102]. Larvae were then dehydrated in a gradient of methanol in phosphate buffer (50, 75 and 100%) and stored at −20 °C in 100% methanol overnight. The next day, samples were rehydrated and permeabilized with 0.04 mg/mL of Proteinase K (Sigma-Aldrich, P2308, Saint Louis MO, USA) for 20 min (dissected brains) or 40 min (whole larvae). Larvae were incubated in blocking solution (IB) with 10% normal goat serum (NGS; Sigma Aldrich, G6767-19B409), 1% dimethyl sulfoxide (DMSO; Panreac Química S.L.U., A3672, Barcelona, Spain) and 0.5% Triton-X-100 in 0.1 M saline phosphate buffer (PBST; pH 7.4) for 1 h at room temperature. After this, larvae were incubated with primary antibody in IB (Tubulin: mouse Anti-Tubulin Acetylated monoclonal antibody, Sigma, T6793, 1:500 dilution; Nrgn: Rabbit Anti-Neurogranin Polyclonal Antibody, Merck, Darmstadt, Germany AB5620, Lot #3091673, 1:500 dilution). The next day, after four washes in PBST (30 min each), samples were incubated in secondary antibody (for Tubulin: Goat anti-Mouse Alexa Fluor 568, Invitrogen, A11004; for Nrgn: Goat Anti-Rabbit IgG-Alexa Fluor 488, Invitrogen, A11008; 1:500 dilution in IB each, Waltham, MA, USA) for one hour at room temperature. After two washes in PBST (30 min each), samples were transferred to 80% glycerol/phosphate buffer (30 min) and mounted in 1% low melting point agarose in 80% glycerol/phosphate buffer for imaging.

Immunoreaction against tubulin (for olfactory organ size) was imaged using an inverted microscope (Olympus CKX53, Barcelona, Spain; 10×; 0.25 NA) equipped with a digital camera (Olympus DP74). Differences in morphology were measured on images using CorelDRAW 2019 design software (v21.0).

Immunoreaction against neurogranin and tubulin was imaged using a laser scanning confocal microscope Nikon A1R equipped with Nikon Plan Fluor 10× (0.30 NA) and 20× (0.50 NA) objectives. Fiji software (https://fiji.sc/; accessed on 1 December 2022) was used to process the confocal z-stacks and to quantify mean fluorescence intensity (measured in Relative Unit Intensity or RUI) [103,104].

### 4.6. Electron Microscopy

The morphology of the olfactory organs was assessed using scanning electron microscopy. Two larvae per condition were fixed at 5 dpf in 2.5% glutaraldehyde at 4 °C overnight. After three PBS washes, samples were post-fixed in 2% osmium tetroxide (Electron Microscopy Science, 19172, Hatfield, PA, USA) for 1–2 h and kept in distilled water. Then, samples were dehydrated, passed through a graded ethanol series and dried by critical-point drying method using CO_2_ (Bal-Tec, CPD 030, Vienna, Austria). Next, samples were covered with platinum–palladium in a sputter coater (Cressington, 208HR, Watford, UK) and examined and photographed in a scanning electron microscope (JSM 6400; JEOL, Tokyo, Japan) equipped with a digital camera (Olympus, Tokyo, Japan).

### 4.7. Gene expression Analysis by RT-qPCR

Total RNA was extracted from pools of 30 zebrafish larvae at 4 dpf, using TRIzol™ reagent as per the recommended protocol (Ambion, 1596026; 500 µL, Waltham, MA, USA). Total RNA was digested with DNase (Promega, M610A, Madison, WI, USA) at 37 °C for 15 min. RNA was purified using the RNeasy MiniKit (Qiagen, Seoul, Korea, 74104) according to the manufacturer’s recommendation. cDNA was synthesized from 1 μg total RNA using GoScript Reverse Transcriptase (Promega, A501C) and OligodT primers (Promega, C110A) with the following thermocycling conditions: 5 min at 25 °C, followed by 1 h at 42 °C and 15 min at 70 °C. Quantitative Reverse Transcription–Polymerase Chain Reaction (RT-qPCR) was performed using GoTaq qPCR Master Mix (Promega, A600A) as per manufacturer’s recommendation with a final volume of 20 µL on a CFX96 Touch System (Bio-Rad) and the following thermocycling conditions: 2 min at 95 °C followed by 40 cycles of 15 s at 95 °C and 1 min at 60 °C. For primer sequences, see Appendix A. Primer efficiency of > 90% was confirmed for each primer pair. Elongation factor 1α (ef1α) was used as a reference gene, and reactions ran in triplicates. The 2^−ΔΔCt^ method was used to determine the relative quantification of gene expression [105].

### 4.8. ICP-MS Analysis of Metal Ions

ICP-MS analysis of zebrafish larvae was performed as previously described [13]. In brief, 10 larvae, anesthetized with MS-222 (4% Tricaine), were pooled and washed several times with distilled H_2_O. Samples were digested in 200 µL concentrated nitric acid at 95 °C until dry and resuspended in 1 mL 3% nitric acid. Further dilution with 20% nitric acid to a final volume of 2 mL was performed prior to analysis. Metals (24Mg, 44Ca and 55Mn) were measured using an Agilent 7500ce ICP-MS instrument with collision cell (in He mode) and Integrated Autosampler (I-AS) using 72Ge as internal standard. The following experimental parameters were used: a) plasma: RF power 1500 W, sampling depth 8.5 mm, carrier gas 0.8 L/min, make-up gas 0.11 L/min; b) quadrupole: mass range 1–250 amu, dwell time 100 msec, replicates 3, integration time 0.1 sec/point. Calibration solutions were prepared for each element between 0 and 200 ng/mL using certified reference standards (Fisher Scientific, Loughborough, UK).

### 4.9. Behavioral Analysis

#### 4.9.1. Experimental Design

Three behavioral paradigms were analyzed at 6 dpf: exploratory behavior in a light or dark environment, light/dark preference assay and optomotor response assay (OMR). For MnCl_2_ exposure conditions, see Appendix A. Only larvae with developed swim bladder were used for behavioral analysis. The customized stimulus protocols and tracking of the freely moving fish were implemented using the Stytra software package (v0.8) [106]. Fish were tracked under infrared illumination (850 nm) with a Mikrotron camera (frame rate of 200 Hz for exploration and light/dark preference assays and 400 Hz for OMR). Custom behavioral analysis was implemented using Python and the bouter package (v0.2.0, https://zenodo.org/records/5931684, accessed on 1 June 2022).

#### 4.9.2. Light/Dark Exploration and Preference Assay

Four fish at a time were placed in a 6 cm Petri dish, and their locomotion was tracked while being presented from below via a projector with a uniform whole-field illumination, a half-field light stimulus (light/dark preference) and a dark stimulus (projector intensity turned to minimum), each consisting of 5 min.

#### 4.9.3. Optomotor Response Assay (OMR)

Individual fish were placed in a 6 cm Petri dish and presented from below via a projector with moving grating stimuli in a closed-loop manner. The time course of the assay consisted of 5 repetitions of rightwards and leftwards moving gratings (20 s each), separated by 10 s of static gratings (pause), for a total time of 5 min per fish. 

### 4.10. Statistical Analyses

Statistical analyses were performed using SPSS 28.0.0.0 (IBM^®^). To compare the two groups, Student’s two-tailed *t*-test was used. For analysis of more than two groups, one-way ANOVA with Tukey or Games–Howell post hoc testing (with or without homogeneity of variance, respectively) was applied. *p*-values (*p*) < 0.05 were considered as statistically different.

## Figures and Tables

**Figure 1 ijms-25-04933-f001:**
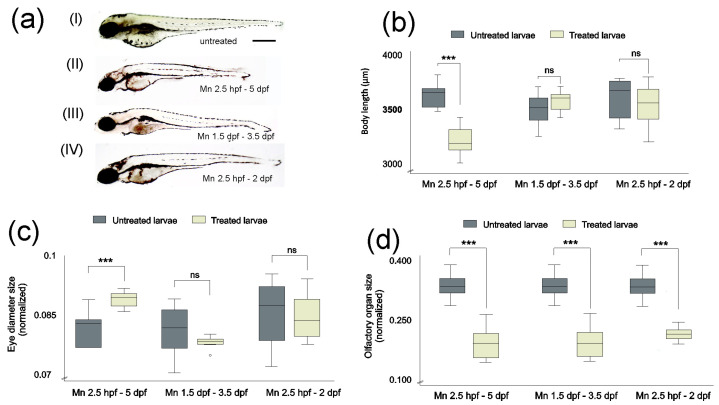
MnCl_2_ exposure causes morphological changes in zebrafish larvae at 5 dpf (**a**(I–IV)), affecting body length (**b**), eye diameter (**c**) and olfactory organ size (**d**). (**a**(I–IV)) Representative images of zebrafish larvae at 5 dpf unexposed (I) and exposed to 500 µM MnCl_2_ at the given durations (II–IV). One-way ANOVA (**b**,**d**) *** *p* < 0.001; (**c**) *** *p* = 0.003; ns. not significant, °correspond to outliers. n = 8 larvae per condition. Scale bar: 100 µm. Data presented as mean ± standard deviation (SD).

**Figure 2 ijms-25-04933-f002:**
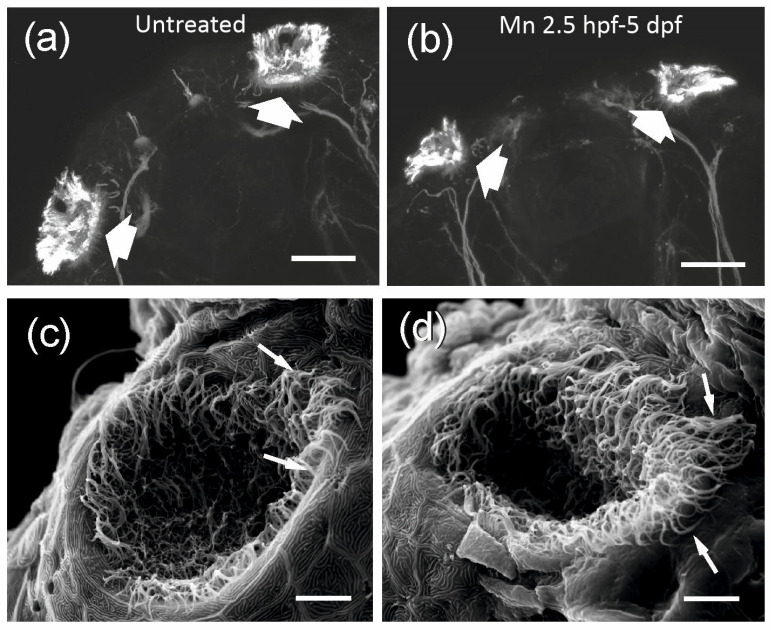
MnCl_2_ alters the morphology of the olfactory organ of larval zebrafish at 5 dpf. (**a**,**b**), Z-projection of confocal images from the rostral top of the head showing the olfactory organ (thick arrows) immunochemically stained against alpha-tubulin in untreated (**a**) and Mn exposed (500 µM, from 2.5 hpf to 5 dpf) 5 dpf larvae (**b**). (**c**,**d**) Scanning electron microscope images of the olfactory epithelium of untreated (**c**) and Mn exposed (500 µM, from 2.5 hpf to 5 dpf) larvae (**d**), showing the morphological changes in the olfactory pit opening and a detailed view of the apical processes (arrows) of the olfactory cells. Scale bars: 50 µm (**a**,**b**); 10µm (**c**,**d**).

**Figure 3 ijms-25-04933-f003:**
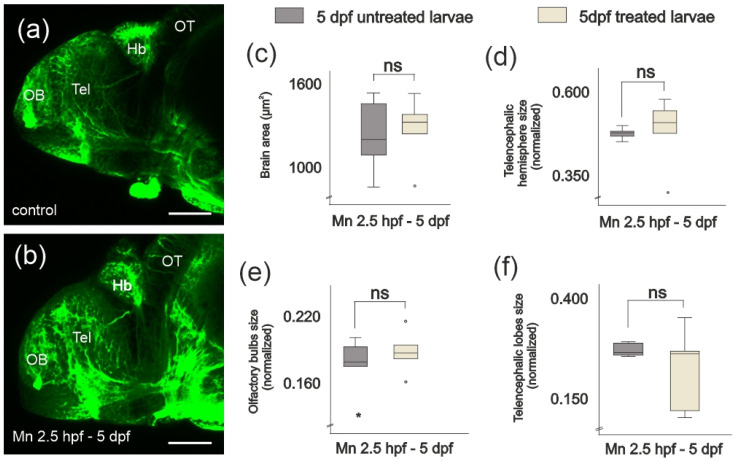
MnCl_2_ exposure does not lead to changes in the area of the lateral profile of the whole brain or the telencephalic regions at 5 dpf. (**a**,**b**) Lateral views of the telencephalon after immunostaining against tubulin in (**a**) negative control larva and (**b**) larva exposed to MnCl_2_ (500 µM). Hb, habenula; OB, olfactory bulb; OT, optic tectum; Tel, telencephalic lobe. Scale bars: 50 µm. Area of (**c**) whole brain, (**d**) telencephalon, (**e**) olfactory bulbs and (**f**) telencephalic lobes in lateral view. The duration of MnCl_2_ exposure (500 µM) was 2.5 hpf to 5 dpf. One-way ANOVA. (**a**) *p* = 0.821; (**b**) *p* = 0.895; (**c**) *p* = 0.410; (**d**) *p* = 0.323. n = 8 larvae per condition. Data presented as mean ± SD. (**d**-**f**) Area normalized to body length. °, * correspond to outliers.

**Figure 4 ijms-25-04933-f004:**
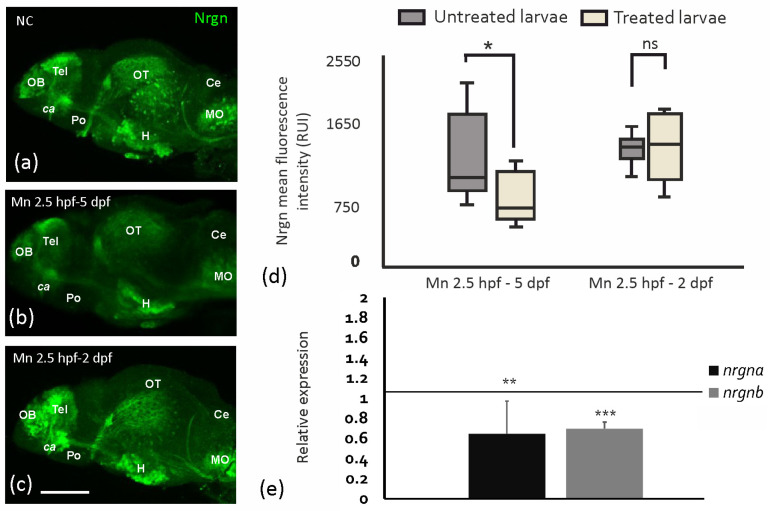
MnCl_2_ exposure reduces Nrgn expression in the brain. (**a**–**c**) Representative images of Nrgn immunostaining of brains at 5 dpf (lateral views). (**a**) Negative control (NC)—unexposed larva. (**b**) Larva exposed to MnCl_2_ (500 µM) from 2.5 hpf to 5 dpf. (**c**) Larva exposed to MnCl_2_ (500 µM) from 2.5 hpf to 2 dpf. ca. anterior commissure. Ce, cerebellum; H, hypothalamus; NC, negative control; MO, medulla oblongata; OB, olfactory bulbs; OT, optic tectum; Po, preoptic area; Tel, telencephalon. Scale bar: 100 µm. (**d**) Neurogranin mean fluorescence intensity levels (Relative Unit Intensity (RUI)) at 5 dpf following MnCl_2_ exposure (500 µM). One-way ANOVA. * *p* = 0.048; ns, not significant. n = 8 larvae per condition. Data presented as mean ± SD. (**e**) mRNA expression of *nrgna* and *nrgnb* at 4 dpf upon MnCl_2_ exposure (500 µM) from 2.5 hpf. Student’s two-tailed *t*-test. *nrgna*: ** *p* = 0.026; *nrgnb*: *** *p* = 5.51 × 10^−5^. n = 6 per condition. Data presented as mean ± SD.

**Figure 5 ijms-25-04933-f005:**
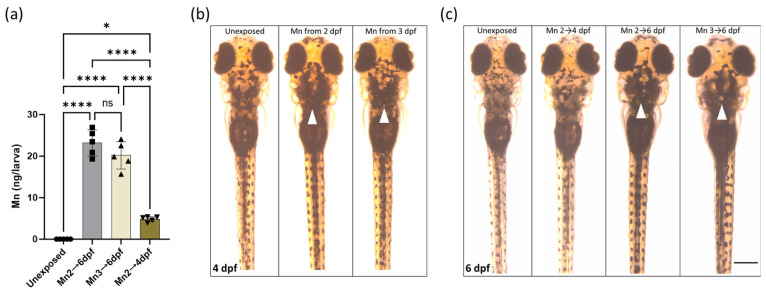
MnCl_2_ exposure causes dose-dependent Mn accumulation in whole larval zebrafish, accompanied by pigmentation changes. (**a**) Mn concentrations at 6 dpf determined by ICP-MS in untreated and MnCl_2_-treated (100 µM) larvae at the given exposure durations. Data are presented as mean ± s.d. (One-way ANOVA with Tukey’s posthoc test; * *p* < 0.05; **** *p* < 0.0001; ns, not significant). (**b**,**c**) Pigmentation pattern of MnCl_2_ exposed zebrafish larvae at the given exposure durations at (**b**) 4 and (**c**) 6 dpf**.** All images are dorsal views of zebrafish larvae. White arrows indicate larvae with abnormal pigmentation compared to control, and hence abnormal visual background adaptation. Scale bar: 250 µm.

**Figure 6 ijms-25-04933-f006:**
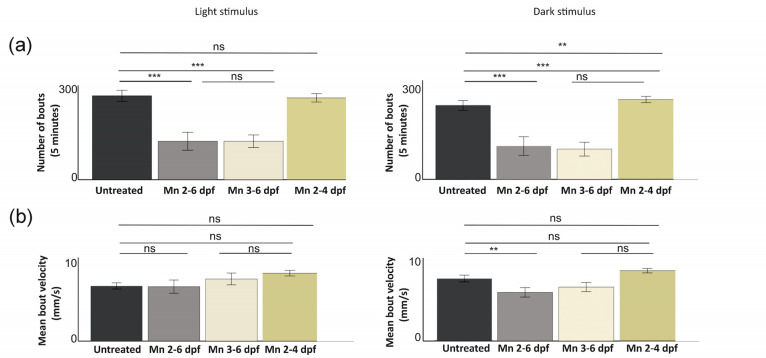
MnCl_2_ exposure effects on zebrafish larval behavior during light and dark conditions, as demonstrated by (**a**) bout number. One-way ANOVA. *** *p* < 0.001; ** *p* = 0.020; (**b**) mean bout velocity (** *p =* 0.020) and (**c**) displacement; ns, not significant. Data presented as mean ± standard error of the mean (SEM); (**d**) fraction of time spent in the illuminated half field, separated in time bins of 1 min each for untreated, Mn 2–6 dpf, Mn 3–6 dpf and Mn 2–4 dpf treated larvae; (**e left**) fraction of correct turns for untreated and MnCl_2_ exposed larvae (when subjected to left- and right-oriented whole field motion stimuli). Bar heights represent the means across all fish in each group; error bars represent the SEM. Only directional bouts (i.e., left- and rightward swims, without forward swims) were considered for quantification. One-way ANOVA is not significant. (**e right**) The number of swim bouts during pause intervals and whole field motion stimuli for unexposed and Mn-exposed larvae. Presented as means ± SEM. Untreated larvae: (**a**–**c**) n = 36; (**d**) n = 68; (**e**) n = 71. Mn 2–6 dpf: (**a**–**d**) n = 32; (**e**) n = 52. Mn 3–6 dpf: (**a**–**d**) n = 36; (**e**) n = 55. Mn 2–4 dpf: (**a**–**d**) n = 64.

## Data Availability

Data are freely available on request.

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
