# Peer review of "Manganese Overexposure Alters Neurogranin Expression and Causes Behavioral Deficits in Larval Zebrafish"

_ijms, 2024, doi:10.3390/ijms25094933_

Round 1

Reviewer 1 Report

Comments and Suggestions for Authors 1、On figture 1, why does each gradient test conduct, separately?
2、On figure 5, are you sure that the p value corresponds to <0.0001? This is hard to believe. As far as I know, the accuracy of these experiments would have been difficult to achieve such values. This needs to be checked carefully.
If needed, please consult a statistician.

3、You may combine some of the figures to reduce the number of figures.
4、The format of the references is inconsistent.

Reviewer 2 Report

Comments and Suggestions for Authors

The authors used a zebrafish model to investigate the effect of manganese overexposure on viability, morphology, behaviour and neurogranin expression levels. The paper is well structured and reads well, but the developmental toxicity of manganese chloride has been studied before and data on the effect of different doses of MnCl2 on zebrafish mortality have been described in several publications. The malformations, behavioural changes and neurotoxicity resulting from manganese intoxication have also been described previously, e.g: DOI: 10.1016/j.aquatox.2016.11.013 or DOI: 10.1016/j.scitotenv.2022.153778.

The authors conclude that manganese does not affect brain size. However, this conclusion is based on the analysis of the lateral view image and the analysis as such is not sufficient to draw conclusions about the size of any 3D object. 

As conclusions in lines 30-32, the authors write that the result shows that Mn overexposure leads to synapse modifications. Neurogranin is not a synapse-specific marker, and even if it was, additional experiments would be needed to conclude that MnCl2 affects synapses. 

The authors mentioned that in Alzheimer's disease, NRGN expression is reduced in the brain, but NRGN levels are increased in cerebrospinal fluid. It would be interesting to know if similar changes occur in zebrafish. Did you see more fluorescent signal (Nrgn immunostaining) in the brain cavities of zebrafish larvae exposed to MnCl2?

A panel of representative images of zebrafish should be added to Figure 3.

Side view images should be added to Figure 5. Supplementary data showing variability in pigmentation between and within groups would be highly appreciated.

 An ethical statement should be added as this research involved procedures on protected animals ( overexposure to MnCl2 after 5dpf).

 What is the explanation for such a large variability between untreated larvae on Figure 4d Mn 2.5 hpf - 5 dpf and why there is no such variability between untreated larvae Mn 2.5 hpf - 2 dpf?

The Materials and Methods section needs improvement:

The authors should clarify: What type of water was used to maintain the zebrafish larvae? Was it tap water, egg water, E3? How were the fish kept? Were they in Petri dishes or multi-well plates? What volume and density was used? At what temperature were the larvae maintained?

Additional notes:

Please specify what dry flake food was used.

How long were the samples permeabilized in ProK?

The description of the electron microscopy should be removed from the Materials and methods as this part is not included in the results.

Nomenclature conventions should be followed when mentioning human and zebrafish genes and proteins.

Round 2

Reviewer 2 Report

Comments and Suggestions for Authors

I am satisfied with the changes